# Feminization of Resistance: Reclaiming the Affective and the Indefinite as Counter-Strategy in Academic Labor Activism

**Aslı Vatansever**

Bard College Berlin, Platanenstraße 28, 13156 Berlin, Germany; a.vatansever@berlin.bard.edu

**Abstract:** 'Feminization' is used either quantitatively to indicate an increased female labor market participation or qualitatively to refer to labor devaluation and to types of work that supposedly require "feminine" skillsets. This article cautiously hews to the qualitative interpretations but suggests an affirmative reconstruction of the concept in the context of collective action. It argues that contemporary grassroots academic labor movements rely more explicitly on collective emotions and aim at building long-term bases of solidarity, instead of performative activism and mass mobilizations. This 'affective turn' in academic labor activism is argued to signal a "feminization of resistance", characterized by a pronounced propensity for affective and relational groundwork. This argument is substantiated in view of the Network for Decent Work in Academia (NGAWiss), a nation-wide precarious researchers' network in Germany, and the New Faculty Majority (NFM), an adjunct advocacy group in the US. The aim is twofold: first, the article contributes to a better understanding of contemporary labor activism by elucidating the precarious collective's incremental achievements, often ignored by the outcome-oriented labor movement literature. Second, by reframing it as a mode of affective resistance, the article extends the analytical scope of the term "feminization".

**Keywords:** academic precarity; affective turn; feminization; labor activism; resistance

## 1. Introduction

As Guy Standing remarked, 'feminization' is an ambiguous term with multiple ironical connotations [1] (P. 583). In recent decades, varying qualitative and quantitative overtones of the concept have found usage in the labor studies literature. The most widely accepted, rather quantitative definition of the term pertains to the demographics of the labor force and refers to the increase in women's participation in the economy, especially from the last quarter of the 20th century onwards [2–4]. As gendered descriptions are anything but value-free, 'feminization' has also come to signify a substantial change in the prevalent employment patterns and organization of work. A second, qualitative, meaning of the term thus focuses on the nature of the work done and the necessary personal traits a worker has to possess for success in a certain job. In this version, 'feminization' basically corresponds to an increased centrality of service and care work in the economy and, subsequently, of supposedly "feminine" qualities required for that type of work, such as empathy, patience, flexibility, and responsiveness [5] (P. 463), [6] (P. 43). The third connotation focuses on the employment status and emphasizes the aspect of labor devaluation by associating 'feminized work' with insecure, low-paid, low-status jobs [7] (P. 46), [8] (P. 3), [9]. In reality, the aspects of work to which these qualitative interpretations of 'feminization' correspond are often interlinked and concomitant as contractually precarious work, especially in the creative/intellectual or service-oriented sectors, usually demands at various levels the workers' affective capacities as well.

Qualitative interpretations of 'feminization' evidently tend to imply pejorative aspects of work and gendered notions of occupational qualifications. Only more recently, the concept has entered the social movements discourse, yet in a rather quantitative sense that refers to instances of collective resistance where female actors numerically constitute the majority or take the lead in the micropolitics of social movement organizations [10,11].

However, an interpretation of 'feminization' that captures the connection between those precarized aspects of work, subsumed under 'feminized labor', and the specific forms of resistance that this type of work engenders remains largely missing from the social movements discourse. Dialectically speaking, the very predispositions that constitute the most exploitable skills in cognitive capitalism may paradoxically offer the most pregnant tools for collective subjectivation. In other words, what makes the workforce individually vulnerable can turn into its collective power. Therefore, research on the political potential of supposedly 'feminized' traits is urgently needed.

This article intends to overcome this gap by proposing a reconceptualization of the term 'feminization' in view of the affective and discursive modes of resistance among precarious researchers. Empirically, it draws on two grassroots academic labor movements—the Network for Decent Work in Academia in Germany (henceforth NGAWiss) and the New Faculty Majority in the US (henceforth NFM) —, which offer salient examples with their processual approach to labor issues in academia and their success in enacting a discursive shift through solidary advocacy work. The choice of subject is motivated by the observation that the academic sector practically features all the qualitative aspects that are associated with 'feminized work', including a noticeable decrease in job security, intersectionality of work and care, an unequal gendered distribution of job insecurity among the workforce, and the prevalence of affective competencies in the labor process [12] (P. 242–243). Therefore, contemporary forms of grassroots labor activism against academic precarity can help us grasp the qualitative implications of 'feminized' work on the nature of the resistance against it.

The theoretical analysis is guided by a dialectic and processual understanding of precarization/feminization, which emphasizes the potential for a new type of subjectivity and novel resistance practices that the loss of securities might bear. Following from this, the main assumption here is that 'feminized labor' (in the negative sense, connoting contractually precarious and emotionally taxing work) generates a specific type of resistance which is also marked by flexibility and indefiniteness and creates a sense of collectivity based on the affective implications of precarity, such as disappointment, anger, and frustration [13]. In addition to advocacy work and discursive critique, grassroots labor activism in feminized sectors often involves a therapeutic–reparative mode of subjectivation based on story-sharing and mutual care. Thus, unlike conventional outcome-oriented forms of labor activism, precarious workers' networks prioritize activities geared towards the formation and solidification of communities of mutual trust and solidarity. This type of community-building effort does not only rely on the rearticulation of work-related grievances as collective causes, but on a pronounced solidary sensibility towards their impact on the actors' personal, relational, and collective well-being. Consequently, while work-related demands may constitute the initial targets, such movements often bear unpredictable outcomes that involve transformations in the fabric of social relationships. This general turn towards more affective, flexible, open-ended, and relational types of collective action, as observed in the context of academic labor activism, is described here as 'feminization of resistance'.

The heteronormative and conventionally gendered predications of 'feminization', even in its reversed connotation outlined above, are of course known to the author. Obviously, the qualitative associations of 'feminization' are closely linked with some assumed characteristics of the 'feminine'. They imply the existence of a "female experiential baggage", indicating certain relational and emotional inclinations [6] (P. 40). Even if the "female experiential baggage" is posited in an affirmative way, perceiving the repertoire of emotional capacities that it entails as fundamentally 'feminine' may seem to conform to conventional gender attributes. However, 'feminization' in this article is not based on an essentialist assumption about an ontological category of the "feminine". Rather, it presents a political consideration of how to politically utilize *the qualities attributed to the historical construct of the "feminine"* The aim is to challenge the patriarchal script of labor activism by depicting how the affective and relational aspects of work and resistance, which have been

de-historicized and denigrated as "feminine" (alongside the female sex to which they were attributed), have come to shape the terrain of work and labor struggles in the 21st century, even in a hitherto male-dominated sector such as academia. By doing so, the article aims at contributing to the discourse on affective forms of resistance.

The article is divided into four parts. The first section following the Introduction elucidates the conceptual framework through which the connection between precarious labor and affective resistance in academia is explicated. The distinctly 'feminized' aspects of academic work are illuminated using the conceptual toolkit of feminist affective approaches to academic precarity. Specifically, the prevalent movement characteristics which are argued to indicate a general tendency of 'feminization of resistance' are explained in conjuncture with the "affective advocacy" framework, as outlined by NFM-activists themselves [14]. The second section explains the methodology and the materials used in the analysis, adopting "participant activism" [15] in the case of NGAWiss and mainly drawing on self-reflective online and printed sources in the case of NFM. The last two sections summarize the findings and their theoretical and empirical implications.

## 2. Conceptual Framework: Exploring the Relationship between Precarized Academic Work and Feminized Resistance

Much has been said on the individual, political, and higher educational implications of casualization in academia, including the rising mental illness rates among researchers, the increased teaching workloads, the deteriorating quality of higher education, the destructive competition, and the loss of collegial solidarity, among others [16–23]. While acknowledging the previously identified individual, social, and political consequences of academic precarity as given, this article slightly shifts the focus. In view of contemporary academic labor movements, it ponders the question of how the affective toll of precarization (i.e., psycho-emotional hyper-flexibility and pronounced attentiveness to external expectations) is being dialectically turned into a strategic advantage within the context of collective action. Located broadly within the discourse on affective forms of resistance, the article thus aims at contributing to a better understanding of the contradictory impacts of precarity on agency.

Using Gallie et al.'s useful distinction between different types of job insecurity, 'feminization' in the context of academic work is understood here as a process that entails both loss of job security in the usual sense (i.e., "job tenure insecurity") and loss of the "valued features of the job", such as autonomy, creativity, and occupational prestige (i.e., "job status insecurity") [24]. According to this, academic work, which has historically been characterized by secure and long-term employment, has now come to be dominated by transience, contingency, and insecurity. Subsequently, "low wages, minimal respect, and temporary positions", which used to characterize women's work in the past, have become the general norm in today's academia [25]. It needs to be noted that, in accordance with the differences in the type and degree of previously existing job security, the process of precarization in the academic sector followed a slightly different trajectory in each context (i.e., direct adjunctification and erosion of tenure in the US as opposed to gradual increase in the number of non-tenured positions at German universities). However, in both contexts, the end result has been in sync with the qualitative connotations of 'feminization', reflected in an overall increase in short-term employment and a subsequent devaluation of academic labor [18,26].

While 'feminization' accurately conveys the aspect of devaluation, this article refers to an ancillary concept to capture the essence of what constitutes the 'feminized' dimension of work. The distinctly 'feminized' traits of the labor process in contemporary academia are partly subsumed here under "maternalized labor", in reference to Fernández-Arrigoitia et al. [27]. With the term "maternalization of adjuncts", Fernández-Arrigoitia et al. refer to a "domestic(ized) role within academia" that involves "performing daily acts of mothering with the students" [27] (P. 95–96). By highlighting the forced care element, the concept of "maternalized labor" complements the notion of 'feminization'. It illustrates the

'affective load'—the main component that actually *feminizes* work: an increased demand on the workers' 'maternal' quality of incessantly attending to others' needs and the exposure to several forms of professional abuse reminiscent of women's historical inferiorization, including *marginalization* in institutional decision-making, *infantilization* as 'early-career', and the *sidelining* as "the 'cheap labor' supporting the 'real' work of the department" [28] (P. 201).

This article identifies two contradictory vectors of 'maternalization' in the context of academic work. In fact, the tendency of feminization within the context of academic labor activism is viewed here as a response to this very tension between the paradoxical courses of 'maternalization'. On the one hand, following Ivancheva et al. [29], the "labor-led contractual precarity" inherent in feminized academic labor is argued to generate a type of "care-led affective precarity", whereby a perpetual cycle of occupational and personal–relational instability occurs when the female faculty especially, with domestic care duties and biological–reproductive concerns, gets excluded from career advancement by the "care-free masculinized ideals of competitive performance" [29] (P. 448). Thus, in addition to the structural decline in job security, a further mechanism of devaluation operates through academic promotion criteria that "reflect gendered norms, favoring hegemonic masculine behaviors", thereby turning the non-tenured female faculty into the "non-citizens of academy" [30] (P. 464–465). As a result, those with domestic care duties or procreant urges are practically being punished for using their 'maternal' capacities for personal/intimate purposes, whereas those who put them completely and unconditionally at the disposal of the academic industry are reprimanded with "affective precarity", i.e., emotional and relational detachment [29] (P. 457).

On the other hand, as much as contractual precarity demands that the non-tenured faculty turn into detached and unemotional hermits with no strings attached, it paradoxically urges them to demonstrate the utmost openness to rapidly changing external demands and an extreme propensity to care and self-sacrifice in their professional lives [31]. In this context, non-tenured staff are expected to approach work-related tasks and relationships with a heightened sense of sensibility, which requires them "to live their professional lives in the 'feminine', i.e., with an increased attunement to life" [32]. This is especially the case in the adjuncts' day-to-day interactions with the students, i.e., the "customers", but certainly not limited to it. Obviously, precarious academics are required to channel the care function that they are forced to withdraw from their personal relationships into their career. In this sense, the 'maternalization' of academic labor seems to proceed along two paradoxical trajectories: a *de-maternalization* in the context of private relationships, accompanied by a *hyper-maternalization* in the professional sphere.

The tension that arises from these contradictory commands of 'de-maternalization vs. hyper-maternalization' is argued here to ignite an emotional reaction on the part of contingent workers, provoking them to retain their professedly 'feminine' capacities through collective action. More precisely, this emotive reaction involves implementing the said interpersonal qualities in a subversive way, i.e., for the purpose of resisting the exploitative forms of feminization in the work context. In a way, it re-cycles the very psycho-emotional skillset subdued by 'feminized work' for activist purposes. Following Doe et al.'s analysis of NFM, this specific form of labor activism that draws on "the potential of emotions to enact change" is described here as "affective advocacy" [14] (P. 226). Contemporary instances of "affective advocacy", as exemplified here by NGAWiss and NFM, are viewed as part of a broader 'affective turn' in labor activism, indicating the rise of a form of labor organizing that involves complex and non-linear relational processes of subjectivation and a less definitive conception of activist 'achievement' (cf. [33,34]). This wider change in the nature and aims of labor activism is described here as the '*feminization* of resistance', denoting a general shift away from linear understandings and strictly rationalist, outcome-oriented forms of labor organizing.

The idea of "affective advocacy" departs from the observation that "emotion and outcry foment dissent"; it signifies an alternative form of activism, whereby collective emotion

is channeled into "real, incremental change" [14] (P. 215). In this context, emotion is directed towards action through listening, sharing stories, self-disclosure, and building long-term bases of solidarity. Considering the slow pace of personalized collectivity-formation, "affective advocacy" often involves time-consuming and unimpressive groundwork geared towards gradual discursive change, rather than spectacular mobilizations or grand transformations [14] (P. 214). Underlying this shift in activist methods and objectives is a more individual life-cycle-oriented view on "precarious employment as traumatic experience that disrupts people's foundational narratives" [14] (P. 230)]. In response to the traumatic subjective impact of precarious labor, labor initiatives tend at times to prioritize the "therapeutic" aspect of activism over the pursuit of "action for a greater cause" [14] (P. 223). Naturally, this requires a change in the idea of 'activist achievement': it challenges the hegemony of large-scale accomplishments, mass mobilizations, and rationalized organizational agency as the sole criteria of a social movement's "success". In view of this turn, this article argues that, to the extent that affective advocacy resists the dominant paradigm of generalizable outcomes and challenges the underlying masculinist doctrine of productivity, it represents a 'feminized' (i.e., counter-masculinist and counter-hegemonic) form of resistance.

The questioning of masculinist understandings of activism should be considered within the context of the increasing inefficiency of traditional forms of labor organization in the face of precarized, fragmented labor. From this perspective, subject-oriented and affect-led labor initiatives can also be seen as a response to the incompatibility of traditional labor organizations with the exigencies of precarious academic labor. At the level of objective material conditions, conventional modes of organizing and mass mobilizations have proven evidently unfeasible in precarious sectors for a variety of reasons, including contractual diversity and legal limitations to disruptive/combative action. In this respect, the difficulties of representing 'feminized' labor through historically male-dominated and performance-oriented forms of labor organization seem to signal a "crisis in masculinized organized labor" [10] (P. 35). Additionally, at the subjective level, conventional labor organizations often fail to address and mend the psycho-emotional implications of a work climate where the workers' own interpersonal skills and empathic dispositions are turned against them. Obviously, any attempt to organize resistance against the exploitation of cognitive and emotional capacities has to start from where the exploitation takes place: the affective dimension of work. Based on these considerations, this article detects a tendency towards feminization in the context of academic labor movements, whereby the "female experiential baggage" [6], i.e., the very set of cognitive capacities that are prone to exploitation within the current regime of accumulation, such as the inclination to care work, adaptability, and responsiveness, is being turned into an advantage for organizing resistance.

## 3. Materials and Methods

This article combines a variety of descriptive and analytical qualitative methods. The insights are drawn mainly from the qualitative content analysis of printed and online sources, including self-descriptive web material, public interviews, and discussion papers by members of both movements, and direct participation in the case of NGAWiss. Following a thematic approach in analyzing the narratives, public interview excerpts and tweets are used with the purpose of detecting common thematic elements across different movements and actions [35] (P. 1).

The common patterns in the data were discerned by using a set of themes and codes derived from the literature on (a) the general connotations of 'feminization' in the context of work and labor movements and (b) the specific features of grassroots initiatives in academia that match up with those connotations. Broader themes included 'the role of collectivized emotions in the subjectivation process', 'gradual change and empowerment as primary objectives', and 'conversational activism as the main mode of action'. Specific codes were more narrowly defined to emphasize the particular characteristics of feminized resistance, such as 'responsiveness', 'organizational elasticity', 'solidary sensibility', and

'counter-hegemonic strategies'. In analyzing the qualitative content, both deductive and inductive approaches were deployed: the former to identify the overall organizational characteristics of NGAWiss and NFM in view of the existing categories of labor activism; the latter to identify the distinctly 'feminized' turn in their aims and activist methods.

In the case of NGAWiss, the author had the advantage of personal access to the Network's activities. Consequently, the analysis is informed by three overlapping roles in the author's own activist-scholarly life cycle:

- Her occupational status as precarious researcher.
- Her activist engagement as a member of the Network's coordinative circle.
- Her research, as part of which she had conducted in-depth interviews with the coordinative circle for another project in 2019 prior to active participation.

By drawing on her own experiences and actively acknowledging the fertile interactivity of her own position and her research, the author transcends the boundaries between being a "scholar-ACTIVIST" (prioritizing activist engagement) and being a "SCHOLAR-activist" (prioritizing the position as researcher) [36]. Thus, in accordance with the counter-hegemonic features of the subject of analysis, a pluralist and non-binary methodological approach is adopted, which is based on the mutual substantiation of activist commitment and knowledge production. More specifically, this article deploys "participant activism" as a way of cross-pollinating research and activism to investigate the case of NGAWiss [15]. "Participant activism" refers to an autoethnographic and reflective methodology that combines 'participant observation' and 'participatory action research' [15] (P. 543–544). It is based on the recognition of multiple "role overlays" between being the researcher, activist, and the subject within the same study, especially in the context of social movement research [15] (P. 546). As such, "participant activism" represents a methodological attempt to overcome the positivistic notion of dispassionate research based on a strict 'object vs. subject' dichotomy.

The analysis of NFM, on the other hand, mainly draws on mission statements and the experiential insights of the members and founders, as expressed in various reflective writings and interviews in printed and online sources. However, as those testimonies and self-reports also reflect a pronounced coupling of scholarly and activist roles as well as a strong emphasis on the validity of subjective experiences, the materials used for the analysis in this article signify an overall 'feminized mode of inquiry'. Underlying this methodological approach is an epistemological premise inherent in feminist narrative research, to which both the author and the NFM-activists seem to adhere: the confidence in the empowering effect of self-disclosure and self-reflexivity and the necessity to "[analyze] both text and experience" as "legitimate sources of knowledge" [14] (P. 216).

## 4. Results

In the following, the two case examples, the Network for Decent Work in Germany and the New Faculty Majority in the US, will be analyzed in view of the guiding themes and codes mentioned previously. The main themes indicating a trend of feminization within the context of academic labor movements include 'affect-led subjectivation', 'slow change and empowerment as main objectives', and 'discursive/conversational activism as the main mode of action'. These correspond to the levels of politicization/organization, goal-setting, and action, respectively. They are supplemented with a set of contextual codes referring to specific characteristics of affective advocacy movements, including *responsiveness*, *organizational flexibility*, *solidary sensibility*, and *counter-hegemonic strategies*. Their manifestations will be traced using two instances: in the case of NGAWiss, the analysis will focus on the latest hashtag-campaign #ichbinHanna that made considerable waves in summer 2021 and resulted in a personal invitation from the Federal Minister of Education and Research for a private audience with the Network's representatives. Among the accomplishments of NFM, the article will focus on the adjuncts' successful push for an increase in the contingent faculty's participation in governance at the American University.

*4.1. "My Name Is Hanna": Self-Disclosure as Subversive Practice*

NGAWiss is a network of precarious researchers, which emerged in early 2017 with the aim to connect the individual local initiatives at different campuses and represent the academic precariat in Germany. Its main target has been the Fixed Term Academic Employment Law (henceforth WissZVG), which has since 2007 limited the so-called 'qualification period' of temporary employment to a total of 12 years (6 years before + 6 years after the completion of the PhD). However, in a cutthroat academic labor market marked by decreased public funding, where 92% of the academic workforce is employed on a temporary basis and sufficient adequate vacancies simply do not exist, WissZVG ended up actually terminating careers [37] (P. 111) [26]. Ignited by this structural impasse, the Network has managed to grow into a nation-wide platform of contingent advocacy within a relatively brief period of time. By now, it comprises over 30 local campus initiatives across Germany, two trade unions, and several federal-level interest groups and disciplinary associations.

The Network is marked by an inclusivist, yet highly diverse, non-membership-based, horizontal structure. Local initiatives as well as individual actors are free to decide on action plans autonomously, while an overarching coordinative circle serves as the main nodal point, maintaining group cohesion and organizational work. Work in the coordinative circle or local initiatives is voluntary and driven by solidarity. Hence, paradoxically, it relies on the extra work and flexibility provided by those who already suffer from unpaid overwork and constant "precarious mobility" between institutions and places in their professional lives [38] (P. 23). The author's own observations and experiences in the coordinative circle confirm that organizational work among precarious researchers often requires not only a great deal of mutual understanding, but also emotional attentiveness to fellow activists' familial obligations, deadlines, frequently changing employment and living situations, health issues, and the like. Nevertheless, as previously stated, engagement in solidary organizational work seems to offer a compensation for the tension between the work-related, forced 'hyper-maternalization' and the concomitant 'de-maternalization' in the private relational sphere. As expressed in the words of a founding member in a reflective discussion paper prior to the official launch of the Network, participation in collective action is seen as a chance for mutual exchange of experiences, solidarity, and the formation of a collective identity as academic precariat [39] (P. 398).

Throughout the last few years, several campaigns have been organized by the overarching Network or its local constituents. Most of the time, protests proceed along conversational lines via press releases, declarations, discussion papers, and the like in social or print media. Even in physical protests, the focus on the social reproductive and psycho-emotional aspects of labor devaluation prevails in the Network's framing of academic precarity. Examples include, among others, 'Frist ist Frust' ('Fixed Term is Frustration'—henceforth FiF), so far the Network's largest nationwide campaign, which started out with a petition for permanent jobs in March 2019 and continued with diverse protest actions across campuses and a very recent sit-in organized by the local initiative at the University of Kassel (KasselUnbefristet), featuring demonstrators with packed suitcases to draw attention to the precarious researchers' constant forced mobility [40,41]. In both cases, the emotional aspects of temporary employment, such as frustration, mental exhaustion, and anxiety over unpredictable prospects, were writ large throughout the protest events and materials.

The same leitmotifs were revisited more strongly through self-disclosing autobiographic notes in the so far most resonant action, the 'Ich bin Hanna' (My name is Hanna) hashtag campaign from June 2021. Preceded by two similarly large and spontaneous hashtag campaigns, #95vsWissZeitVG and #ACertainDegreeOfFlexibility, #IchBinHanna was kicked off impulsively by three individual NGAWiss members in response to an animated propaganda video published by the Federal Ministry of Education and Research. The original video from 2018 featured a cartoon early-career researcher named Hanna, who was seen in the video happily defending temporary contracts as a way to prevent "one generation of researchers from congesting all positions". Following the three members' initial tweets in June 2021, within a few days hundreds of early- and mid-career researchers

with temporary positions joined in a twitter storm, where they flooded the newsfeed with personal short biographies revealing heart-wrenching details of life as a precarious academic under the hashtag #IchBinHanna. Within only 4 days, the hashtag featured in over 45,000 tweets [42]. The rapid snowballing of tweets demonstrated the significance of 'empowerment' as an objective in affective advocacy, proving that "resistance inspires, provokes, generates, encourages ( . . . ) resistance" [43] (P. 52).

Most entries were marked by pronounced sarcasm but also anger over the Ministry's attempt to gaslight precarious researchers into celebrating their own precarity as a necessary instrument of scientific innovation [44]. As expressed by one of the initiators, Kristin Eichhorn, in a public interview, the main aim was to reveal the unethical and degrading approach underlying the Ministry's view on academic workers as "items"—and not humans—that "congest" the system and to demonstrate "the individual fates hidden behind the phenomenon [of temporary employment]" [45]. In other words, the protestors wanted to demonstrate against the neoliberal paradigm of generalizability, that they are in fact "not a bargaining chip that is supposed to make the system more flexible; but human beings with families and dreams, who have a right to decent working conditions" ([42] quoting one of the initiators). In this spirit, non-tenured researchers tweeted their academic backgrounds and current employment status, often revealing their personal, work-affected psycho-emotional state. Through self-disclosure, the protestors thus subverted de-humanized portrayals of temporary academic work that were put in circulation by the academic and political establishment:

- Twitter Activist1 _ Oct 23

The fact that I'm losing sleep, because I'm worried not to be able to cope with the incessant enormous workload anymore, hasn't changed a bit throughout these 10 years in academia. My concerns feel like existential. And they are existential. Because: #IchBinHanna [translated from the German original—A.V.]

- Twitter Activist2 _ Oct 18

#IchBinHanna. Rejection letter for a professorship in Germany I couldn't even remember I had applied for—that's how long ago.

- Twitter Activist3 _ Jul 9

It's not only the fear & anguish of hoping for another gig the following year. What's often forgotten is the effort & expenses it takes to move every other year for another Postdoc! When we will move this September, we'll have lived in 4 countries in 3y.#IchBinHanna Europe-wide.

- Twitter Activist4 _ Jul 15

#IchBinHanna #IchBinHannaCH My second daughter was born while I was working as a scientific collaborator at a University of Applied Sciences and Arts. It was a part-time and short-term job.

- Twitter Activist5 _ Jul 29

When people say, 'you can continue to do research whatever your job is, be an independent scholar, if you love it so much', I know it's well-meaning, but it really grates on me. Research is work and labor. Why and how would I do this if I'm not being paid?! #IchBinHanna.

- Twitter Activist6 _ Jul 20

Retirement? Having worked multiple fixed-term academic jobs in 3 different countries, there won't be much pension for me. I'd better sooner rather than later turn into a pure spirit without any mundane needs. Most appropriate for a philosopher anyway. #IchBinHanna #Altersarmut.

- Twitter Activist7 _ Jun 10

XYZ, 42, Anglicist, #firstgen. 0 children (why, possibly?). Doing research on history of obstetrics. 8 contracts, 12 years are over. Love research & teaching, esp. students. Sleep terribly at night and have panic attacks. Congesting the system with third-party-funding. Who are you? #IchBinHanna [translated from the German original—A.V.].

- Twitter Activist8 _ Jun 10

The Federal Ministry of Education and Research lets temporarily employed researchers perish and mocks them on top of that. To remind you of the fact that WissZVG operates against human beings, I'm hereby giving the academic precariat a face: #IchBinHanna [transl.—A.V.].

Upon widespread critique, the Federal Ministry of Education and Research has removed the video, as can be seen under its former link https://www.bmbf.de/de/media-video-16944.html (accessed on 10 October 2021). Additionally, as a result of the unexpected twitter storm, the Minister of Education and Research, Anja Karliczek, saw herself compelled to express the need for more permanent positions in academia [46]. The change in tone was followed by a plenary session of the German parliament in June 2021 and a personal invitation from the Minister to the coordinative circle of NGAWiss for a closed audience in late summer 2021 [47]. Practical outcomes in terms of an increase in permanent employment at the universities still remain missing. Nevertheless, through powerful personal testimonies and story-sharing, #IchBinHanna managed to channel work-related emotions into a collective affect-led action and contributed immensely to the ongoing transformation of the discourse on academic employment relations.

In addition to the political and discursive resonance of single campaigns, affective advocacy generally fosters collegial relationships along the way—something that can be viewed as essentially counter-hegemonic in neoliberal academia where work-related issues are rationalized to the degree of de-humanization and collegial solidarity is being eroded by hyper-competition. According to another initiator of the #IchBinHanna campaign, Amrei Bahr, #IchBinHanna itself was born out of virtual camaraderie and emerged "through engagement" in conversational activism. In this sense, the mostly invisible processual (i.e., *feminized*) groundwork that NGAWiss and its local constituents have been doing for the last few years has evidently contributed to the counter-masculinist transformation of the relational infrastructure in German academia. As the case of #IchBinHanna (and NGAWiss in general) indicates, 'organizational flexibility' is conducive to agential creativity and allows for more personalized affective reactions to emerge and resonate collectively. In the meantime, the way the Network members instantly close ranks and support each other by sharing their own stories at the risk of making themselves vulnerable demonstrates how the ethics of 'solidary sensibility' has taken roots in the precarious researchers' grassroots initiatives, reverting the damage of precarious work and hyper-competition on esprit de corps in academia.

*4.2. Planting Roots in Quicksand: Contingent Faculty Builds Communities*

As the name of the movement suggests, the New Faculty Majority emerged as the advocate of the non-tenured 'new faculty majority" with no contractual or legal job security, whose number has increased by 259% among full-time and 286% among part-time faculty between 1979 and 2011 in the US [48] (P. 236). Starting out in listserv around 2009, NFM managed to create a "real community" and now consists of two affiliated organizations, namely the National Coalition for Adjunct and Contingent Equity (non-profit membership organization) and the NFM Foundation (public charity) (http://www.newfacultymajority.info/history-and-foundational-principles/. accessed on 20 October 2021).

Attesting to the catalyzing role of affect in feminized resistance, the former president and executive director of the NFM Foundation, Maria Maisto, points to "anger" as the main trigger in the emergence of NFM—anger over the silencing of contingent faculty and over tenured colleagues' "ignorance", "denial", "sexism", "hostility", and "resistance to thinking about the work that we were doing in the context of the labor movement and to think about academic work as work" [49]. The same reluctance to treat academic work

as a form of wage labor was also partly the reason for the slow unionization rates and the strong anti-union language in US academia despite the striking decrease in tenure since late 1970s [50] (P. 8). Under these circumstances, NFM adopted strategies that shifted away from conventional labor organizing and geared towards "consciousness raising", "advocacy", and "resource sharing" with the explicit aim of challenging the hegemonic paradigm of academic exceptionalism [51].

A major obstacle to consciousness raising pertains to the autocratic structures of university governance. The exclusion of temporarily employed staff from decision-making in matters related to hiring practices, fund management, and curriculum development has been identified by NFM activists as an effective way to silence precarious voices in the university context [52] (P. 183). The undemocratic practices of university governance are often buttressed by a general inferiorization of non-tenured faculty and a depreciation of their professional expertise due to their employment status [53] (P. 151). To make things worse, non-tenured faculty typically lacks the institutional stability required for proving one's value as an academic and for establishing long-term relationships of solidarity with peers. However, the steady growth in the full-time contingent faculty over the last decades has not only started to undermine the status of tenure as the sole criterion of academic merit but has also paradoxically led to a relative solidification of adjunct communities within and between institutions. Influenced by the efforts of adjunct advocacy organizations such as New Faculty Majority, Faculty Forward, and PrecariCorps, contingent workers have managed to push forth some incremental progress in terms of the democratization of university governance.

The case of the "increasingly stable population of contingent faculty" at the American University (henceforth AU), who managed to strike roots in their respective departments following the introduction of 5-year contracts in 2009, represents a salient example [28] (P. 201). With over 40% of its academic staff consisting of part-time adjuncts and over 20% of full-time fixed-term staff ("term faculty"), the AU faced a governance crisis from 2005 onwards. Subsequently, under increased pressure from adjunct advocacy groups to change its top-down mode of administration, the institution underwent a drastic governmental change in favor of the contingent faculty's extended participation in decision-making [54]. According to the activists who took part in the long and arduous process of adjunct advocacy at the AU, "these changes have resulted from collaboration between tenure-track and contingent faculty—but that collaboration itself resulted from the efforts of American's contingent faculty" [28] (P. 199). In this respect, the adjuncts' struggle to democratize university governance followed a counter-hegemonic strategy in two senses: first, in the absence of established organizational structures of labor representation, the contingent faculty had to rely on their own resources and collegial solidary sensibilities to enact change. Second, effective advocacy needed to challenge and overcome the long-established dichotomies between tenured faculty ("real department") and the non-tenured staff ("cheap complementary labor") [28] (P. 201).

The formation of communities in institutions, or "stabilization" as Maria Maisto describes it, requires a steady effort to gain institutional standing and build alliances with tenure-line staff. In short, it requires routine and relatively unimpressive relational infrastructural work. The case of AU provides a live example: adjunct activists in the writing department of the AU emphasize that the "stabilization" process involved long efforts on their part to overcome their own "timidity", which had resulted in the first place from their systematic infantilization vis-à-vis their tenured peers [28] (P. 200). To make their demands heard and raise consciousness, adjunct activists had to gain "faculty reputation" as equals in academic merit; this required overcoming the internalized sense of inferiority and incompetence in the first place [28] (Pp. 201–202).

Obviously, the first and most crucial step in adjunct advocacy involves addressing and surmounting the psycho-emotional impact of devaluation, i.e., 'empowerment work'. As Eileen Schell remarks, "solidarity can be created, in part, through tenure-track and non-tenure track faculty working in concert with one another to change working conditions.

However, working conditions are tied to complex emotional states and life narratives that must be acknowledged as well—the anger, disappointment, frustration, fear, anxiety, timidity and other emotions that both galvanize and halt action" [55] (Pp. xii–xiii). In short, effective adjunct advocacy requires first and foremost an attempt to revert the psycho-emotional damage of precarity and devaluation, followed by a deep and enduring restructuration of peer relationships around a restored sense of value and collegiality. One of the handicaps the NFM activists draw attention to is that this type of slow and plodding relational groundwork often goes unnoticed. This is mostly due to the fact that it usually yields only humble wins, such as access to faculty boards, positive change in the attitude of tenured colleagues, creation of useful alliances, or adjustments in curriculum and teaching load, which the productivity-oriented masculinist approach to social movements could easily dismiss as not significant enough [14] (P. 214). However, these incremental victories gain importance once we realize that the long-term, inspiring imprints of these struggles and discursive shifts on the institutional memory are actually bigger than their contextual outcomes.

## 5. Discussion

The analysis identifies a tendency of 'feminization' at two interrelated levels in grass-roots labor movements in the academic sector: (a) at the level of organization and subjectivation and (b) at the level of tactics and outcomes.

With regard to the former, the analysis demonstrates that academic labor initiatives do not conform to conventional structures of labor organizing. Instead, they are marked by a simultaneousness of the moment of subjectivation and organization. This has basically two implications in terms of the feminization of resistance: on the one hand, the absence of a hierarchical, professionalized labor organization organically subverts the dichotomies between "organizing cadres vs. activists", "leadership vs. follower", or "liberator vs. liberated", which are based on an idea of "rationalized, impassionate (i.e., strategic) leadership" as opposed to what is thought of as "spur-of-the-moment, impulsive insurgence". Practically, academic labor initiatives transcend these organizational hierarchies with their organizational elasticity and emphasis on individual agency.

On the other hand, in a less trivial sense, the overlapping of subjectivation and organization suggests the removal of the duality that the masculinist paradigm of resistance had inserted between politics and emotions as well. At a deeper level, underlying the abovementioned conventional understanding of organized labor activism are Cartesian dualities such as 'rationality vs. emotionality', 'politics vs. emotions', and, ultimately, 'mind vs. body'. This binary compartmentalization of human capacity notoriously delegitimizes emotions in the context of political agency and has been widely criticized by feminist, queer, and postcolonial studies [56] (P. 583). Grassroots labor initiatives of precarious researchers necessarily re-integrate the 'emotional' into the 'political', as the emotional implications of work are often at the root of both individual politicization and the formation of solidary workplace communities. Subsequently, adjunct advocacy is generally marked by a heightened attentiveness towards the subjective impacts of precarity, which are usually not in the radar of traditional labor organizations. As the analysis shows, this is the case in NGAWiss' constant emphasis on social reproductive concerns, health issues and anxiety; it has also been the case in NFM activists' self-reflective efforts to overcome their own work-status-related "timidity" [28]. Thus, in affective advocacy, 'emancipatory agency' and 'emancipatory knowledge' are not only congregated in the same (collective or individual) body, but they are first and foremost rooted in 'emancipatory emotions', such as *rage* over devaluation and inferiorization, *disappointment* over bleak career prospects despite hard work, and *fear* and *anxiety* over an increasingly ambiguous future.

The central role of emotions, perceptions, and experiential factors in the subjectivation process also shapes to a great extent activist tactics and the actors' assessment of achievements. Accordingly, the repertoire of actions in precarious initiatives seems to depart from the feminist premise of "the essential validity of personal experience" [57] (P. 119). The

findings indicate the frequent collectivization of work-related emotions as a central strategy in precarious researchers' initiatives. Moreover, as the case of #IchBinHanna demonstrates, action campaigns often reflect an "ethic of caring through sharing", whereby actors support each other by sharing experiences and expressing vulnerabilities as a way of protecting each other from stigma and other possible repercussions of activism [58] (P. 778). Resisting the possible individual consequences of political action by collectivizing vulnerability represents a radical break from the neoliberal "individualization of misery" [59]. In this sense, the *de-individualization of hyper-individualized structural problems* actually represents the main achievement of affective advocacy. In other words, the collectivization of emotions and vulnerabilities is not only a tactic in affective advocacy; for feminized modes of resistance, it is an aim in itself. This highly responsive and care-oriented form of activism continues to challenge both the existing employment relations in academia and our understanding of labor activism.

However, two concerns may arise in the face of the organizational amorphousness and the indefiniteness of outcomes prevalent in feminized labor organizing. One is related to the durability of the movements themselves; the other concerns the analytical parameters of what counts as resistance. First, single social movement organizations in the horizontal and affective (i.e., feminized) end of the spectrum are likely to be short-lived. Flexible collectives whose stamina and dynamism are based solely on volunteers' capacities, willingness, and life situations possibly run the risk of disbanding in the face of changing circumstances. As Berry and Worthen remark in a recently published comprehensive account of contingent organizing in the US, the lack of structure and continuity in grassroots academic labor activism is probably "partly the consequence of the very temporary-ness—contingency—of [their members'] employment" [60] (P. 144). However, at the same time, this flexibility is also likely to allow for the emergence of other initiatives when certain collectives dissolve, making sure that "the movement moves on" and "what is permanent is the wave itself" [60] (P. 145).

Second, too much emphasis on the subjective and emotional may lead to a self-celebratory view on resistance and end up including every expression of work-related discontent in the category of labor activism. By doing so, it may inadvertently contribute to the discrediting of the real infrastructural work done by active contingent advocacy groups and networks. Moreover, the standard question, whether anti-precarity initiatives, which are loosely organized along the blurry line between solidarity and resistance, qualify as 'labor activism' needs to be addressed. As compared to more resolute forms of workers' organizations of the past, today's collectives are usually disregarded by the masculinist notion of social movements. They are often viewed as "communities of coping" [61] (P. 58) or "peg communities" [59], that serve as informal forums to cope with the psychological burden of precarity. As such, they appear to be nothing more than "a momentary gathering around a nail on which many solitary individuals hang their solitary individual fears" [59] (P. 37).

It is true that networks and solidarity initiatives usually do not possess the capacity to organize confrontational forms of action such as strikes. In this sense, the discursive success of grassroots academic initiatives in challenging the 'tenure-normative' world of academic work should not lead us to "[overestimate] the reparative mode of affective politics" and "celebrate affective politics as a new, all-encompassing form of politics" [56] (P. 584). After all, amidst the ongoing discursive shift, precarious researchers are still confronted with deteriorating working conditions—and their improvement requires additional leverage. Nevertheless, as the main hypothesis suggests, affective advocacy does not only compensate for the lack of unionized struggles in the practical sense but represents a challenge to this hegemonic masculinist paradigm of productivity at the level of both labor organizing and academic production.

## 6. Conclusions

The main aim of this article was to extend the analytical scope of the term 'feminization'. This has been achieved basically in three steps:

First, by carving out the contradictory vectors of precarization as 'de-maternalization in the private sphere' and 'hyper-maternalization in the workplace', the analysis has demonstrated what exactly constitutes the distinctly "feminized" character of precarious academic work. This insight helped illuminate both the psycho-emotional triggers of contention and the presumably "feminine" affective skills that are somewhat developed forcefully within the labor process and are later paradoxically utilized in resistance practices.

Second, following from there, the article established the relationship between precarious academic work and the specific form of grassroots labor activism that it inspires. More precisely, the findings confirmed the presumed role of affect in reviving political agency and creating a new type of labor activist collectivity—one that emerges from (and acts upon) shared emotions and organizes around the principles of mutual care and story-sharing.

Third, by demonstrating that the very 'feminine' qualities which enable exploitation in the workplace can dialectically turn into a source of collective power in the context of resistance, the analysis resuscitated affirmative connotations of the historical construct of 'feminine'. The insights drawn from personal testimonies and activist inputs of both movements' actors confirmed the initial hypothesis that qualities ascribed to the 'feminine', such as vulnerability, attentiveness, flexibility, and propensity to care, indeed constitute the core of collectivity building in affective advocacy. This, as the analysis highlighted, also marks the particularity of contemporary grassroots academic labor activism vis-à-vis the conventional forms of organized labor, which are characterized by an outcome- and mobilization-oriented mode of activism and are therefore described as 'masculinist'. Acknowledging this contrast, the article illuminated how the current academic labor initiatives represent a shift away from the latter. Thus, it did not only detect an affective turn in current resistance practices but expanded the exploratory range of the term 'feminization' by associating it with counter-hegemonic modes of labor activism and demonstrating how 'feminized' intersubjective qualities provide an advantage in contemporary resistance practices—one which the traditional labor organizations fail to offer in the face of a fragmented labor force.

As Baaz et al. contend, "the fundamental and possible normative value of resistance" lies in its capacity to "extend the space for making choices and open up possibilities by undermining, destabilizing, or restructuring such power relations that limit and produce our (possible) identities, actions, space or bodies" [62] (P: 34). This article analyzed contemporary academic labor struggles in the light of a revised notion of feminization and highlighted the potential of affective strategies for extending the said space for agency. This humble step will hopefully inform future research and allow for a better understanding of the political potential of allegedly 'feminine' traits in grassroots labor organizing.

**Funding:** This research was funded by the Philipp-Schwartz-Initiative of the Alexander von Humboldt Foundation.

**Institutional Review Board Statement:** Not Applicable.

**Informed Consent Statement:** Not Applicable.

**Data Availability Statement:** The data presented in this study are available on request from the corresponding author.

**Conflicts of Interest:** The authors declare no conflict of interest.

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
