# Peer review of "Feminization of Resistance: Reclaiming the Affective and the Indefinite as Counter-Strategy in Academic Labor Activism"

_publications, doi:10.3390/publications10010001_

Round 1
Reviewer 1 Report
Focused on the phenomenon that the author conceptualizes as the ‘feminization of resistance’, the article examines resistance and labor activism in academia and its apparent tendency to rely more explicitly on collective emotions and aim at building long-term bases of solidarity (the feminine) than traditional forms of resistance such as performative activism and mass mobilizations (the masculine). Herself navigating the positions of both a scholar-ACTIVIST and a SCHOLAR-activist, the author adopts the auto-ethnographic and reflective methodology of ‘participant activism’, which combines elements of participant observation with elements of participatory action research. Through deploying descriptive and analytical qualitative methods and a combination of deductive and inductive approaches, the author zooms in on two recent instances of academic labor resistance to develop and substantiate her argument: the Network for Decent Work in Academia (Germany) and the New Faculty Majority (US). The analysis reveals an ‘affective turn’ in academic labor activism, which the article argues signals feminization of resistance.
This is an excellent article on all counts. It picks up an extremely topical and relevant issue and does so with great motivation, respect for and commitment to the cause studied and the (vulnerable) group behind it. It does a fantastic job at identifying, bringing to light and making sense of what is – as the author herself acknowledges – largely ‘invisible’ and ‘unimpressive’ but extremely labor-intensive, important and indeed powerful affective and relational groundwork carried out by people in the academic communities. It masterfully and compellingly conceptualizes and theorizes the observed resistance practices of affective advocacy, linking those to the big debates in the feminist scholarship, scholarship on contemporary academia and on labor resistance/activism/movements. It is well-structured and well-argued. It is written in an eloquent manner.
The article will be of interest – and is indeed a must-read – for any individual working in or on contemporary academia, be it as a scholar, or an higher education administrator/manager, or a policy-maker.
I do not have any immediate suggestions on how the author might improve the article.
Author Response
I thank Reviewer 1 very much for her/his/their encouraging and appreciative comments on my article and for advising its publication.
Reviewer 2 Report
An inspiring text that gave me a lot of new food for thought. Very interesting is the implicit question of the extent to which the new work relationships in science also create a feminized sector ("feminized work"/"maternalized labor") - similar to other professionalization processes, e.g. in medicine, where there has been an allocation of assistant work to women and a feminization and outsourcing of certain activities (such as nursing) from the profession. I would recommend the publication, after a slight revision that compensates for some (rather technical) weaknesses:
It should be made clearer, or announced more clearly at the beginning, that the article addresses both the feminization of scientific work and the potential of feminization for political resistance - and how the two are interrelated.
Also desirable would be a clearer connection to a state of research - be it related to "affective forms of resistance" or "grass-roots labor movements" "affective labor" or similar (just examples). This doesn't have to be done in a very extended way, it should just show what state of research the article contributes to.
Other:
"academic work, which has historically been characterized by secure and long-term employment, has now come to be dominated by transience, contingency, and insecurity."
=> This is not true, at least it does not apply to Germany - in this respect, a differentiation would have to be made here between working conditions in U.S. scientific organizations and German universities/research institutes.
Secure and long-term work was today as in former times in Germany only for scientists with civil servant status. The title of professor is decisive here, because all other scientific activities have always been designed only for temporary employment. Precariousness at this level is therefore nothing new; what is new is that there are now more of these positions than of the secure ones.
"the materials used for the analysis in this article signify an overall ‘feminized mode of inquiry’. Underlying this methodological approach is an epistemological premise inherent to feminist narrative research"
=> the methods are thus also "feminized"? At this point at the latest, it must become clear what is actually meant by the concept of "feminization" and to what extent the term is used as a concept in this article.
In particular, it should be clarified how "self-disclosure" is related to the feminization of protest forms. Is it the modes of self-disclosure? By referring to personal fates rather than telling success stories?
Keyword "anonymization" and "protection of research participants": Are the Twitter activists (Amrei Bahr @AmreiBahr etc) informed that their names will be published as part of the article?
Apart from the Twitter quotes, no empirical data is used. This does not require a "feminized mode of inquiry", which is why it does not really make sense to announce it. In addition, the contributions are only analyzed in a rudimentary way.
The final chapter lacks a conclusion regarding the promised goal of the article to extend the analytical scope of the term "feminization".
Reviewer 3 Report
The manuscript is an interesting research and the subject is relevant and appropriate for the scientific journal.
The purpose is clearly stated and literature review summarizes many studies regarding the topic. The author should write his critical opinion about the literature on the field.
Research methodology, results and discussion are clearly presented. The conclusions should be presented in a different section and should be improved.
Formal requirements are not respected.
Author Response
I thank Reviewer 3 for their comments. I have now added a separate Conclusion.
However, I do not quite understand in what way the "formal requirements are not respected", as I followed the submission and referencing guidelines precisely. I would appreciate if Reviewer 3 could elaborate on that and point to concrete instances where the formal requirements were not respected.